# The Speed of Ingestion of a Sugary Beverage Has an Effect on the Acute Metabolic Response to Fructose

**DOI:** 10.3390/nu13061916

**Published:** 2021-06-02

**Authors:** Mehmet Kanbay, Begum Guler, Lale A. Ertuglu, Tuncay Dagel, Baris Afsar, Said Incir, Arzu Baygul, Adrian Covic, Ana Andres-Hernando, Laura Gabriela Sánchez-Lozada, Miguel A. Lanaspa, Richard J. Johnson

**Affiliations:** 1Division of Nephrology, Department of Medicine, Koc University School of Medicine, Istanbul 34010, Turkey; tdagel@kuh.ku.edu.tr; 2Department of Medicine, Koc University School of Medicine, Istanbul 34450, Turkey; bguler@kuh.ku.edu.tr (B.G.); lertuglu14@ku.edu.tr (L.A.E.); 3Division of Nephrology, Department of Internal Medicine, Suleyman Demirel University School of Medicine, Isparta 32260, Turkey; afsarbrs@yahoo.com; 4Department of Biochemistry, Koc University School of Medicine, Istanbul 34010, Turkey; sincir@kuh.ku.edu.tr; 5Department of Bioistastics, Koc University School of Medicine, Istanbul 34010, Turkey; abaygul@ku.edu.tr; 6Department of Nephrology, Grigore T. Popa’ University of Medicine, 700115 Iasi, Romania; accovic@gmail.com; 7Division of Renal Diseases and Hypertension, University of Colorado Denver, Aurora, CO 80045, USA; ana.andreshernando@cuanschutz.edu (A.A.-H.); miguel.lanaspagarcia@cuanschutz.edu (M.A.L.); richard.johnson@cuanschutz.edu (R.J.J.); 8Department of Cardio-Renal Physiopathology, Instituto Nacional de Cardiología-Ignacio Chavez, Mexico City, Mexico; lgsanchezlozada@gmail.com

**Keywords:** sugary beverage, fibroblast growth factor 21, osmolarity, copeptin

## Abstract

Background: The consumption of sweetened beverages is associated with increased risk of metabolic syndrome, cardiovascular disease, and type 2 diabetes mellitus. Objective: We hypothesized that the metabolic effects of fructose in sugary beverages might be modulated by the speed of ingestion in addition to the overall amount. Design: Thirty healthy subjects free of any disease and medication were recruited into two groups. After overnight fasting, subjects in group 1 drank 500 mL of apple juice over an hour by drinking 125 mL every 15 min, while subjects in group 2 drank 500 mL of apple juice over 5 min. Blood samples were collected at time zero and 15, 30, 60, and 120 min after ingestion to be analyzed for serum glucose, insulin, homeostatic model assessment (HOMA-IR) score, fibroblast growth factor 21, copeptin, osmolarity, sodium, blood urea nitrogen (BUN), lactate, uric acid, and phosphate levels. Results: Serum glucose, insulin, HOMA-IR, fibroblast growth factor 21, copeptin, osmolarity, sodium, BUN, and lactate levels increased following apple juice ingestion. The increases were greater in the fast-drinking group, which were more significant after 15 min and 30 min compared to baseline. The changes in uric acid were not statistically different between the groups. Phosphate levels significantly increased only in the fast-drinking group. Conclusion: Fast ingestion of 100% apple juice causes a significantly greater metabolic response, which may be associated with negative long-term outcomes. Our findings suggest that the rate of ingestion must be considered when evaluating the metabolic impacts of sweetened beverage consumption.

## 1. Introduction

As much as 20% of daily total energy intake is through caloric beverages, according to the United States Department of Agriculture (USDA) [1]. Among youth, soft drinks and 100% fruit juice, which have high sugar and caloric contents, account for more than one-fifth of total beverage intake [1]. However, mounting evidence indicates that heavy consumption of soft drinks contributes to the development of obesity, increased risk of cardiovascular disease (CVD), and type 2 diabetes mellitus (T2DM) [2,3]. Notably, the European Prospective Investigation into Cancer and Nutrition (EPIC) cohort reported a positive association between consumption of soft drinks and all-cause mortality [4]. 

Fructose is the main sugar found in both 100% fruit juice and sugar-sweetened beverages [5] and is responsible of many of the health consequences associated with sweetened beverages, including insulin resistance, dyslipidemia, and visceral obesity [6]. Fructose can vary in percentage from 50% to 65% of sugar content in sugar-sweetened beverages and up to 67% in some 100% fruit juices, which is up to 10 times more than that present in a whole fruit. Contrary to common belief, the fructose contents of 100% fruit juices are equal to or even higher than most sugar-sweetened beverages, reaching concentrations of 67 g/L in some commercially available 100% apple juices [5]. Although 100% fruit juice is usually perceived as a healthy beverage and a rich supply of vitamins and phytonutrients, it is not the same as whole fruit since fiber content is lost while sugar content is considerably higher.

Although fructose and glucose have the same chemical formula, fructose metabolism is independent of insulin and does not have hormonal regulation. Fructose generates more pyruvate and induces gluconeogenesis, glycogen accumulation, and lipogenesis. Recent studies suggest that, after ingestion in low doses, fructose is primarily metabolized in the small intestines, while high doses saturate the absorption of fructose in the intestine, which causes higher amounts of fructose to be carried and metabolized in the liver [7], thereby triggering hepatic lipid accumulation [7,8]. The total increased fatty acid formation may contribute to weight gain [9], while intrahepatic lipid accumulation and fructose-dependent oxidative stress are believed to induce nonalcoholic steatohepatitis [8,10]. A high fructose intake has been consistently linked to metabolic syndrome both in humans [11] and in animal models [12,13]. 

One aspect of sugary beverages is that they are not only high in fructose content, but also typically ingested rapidly. Studies have shown that the mechanism via which fructose induces metabolic syndrome is not dependent on the caloric content of fructose, but rather due to the unique ability of fructose to cause intracellular adenosine triphosphate (ATP) depletion and activate an adenine nucleotide degradation pathway in the liver [8,14,15,16]. The effect of inducing ATP depletion is more severe with a higher concentration of fructose [8]. This raises the possibility that the speed of ingestion might be an important risk factor for developing metabolic effects of fructose. Indeed, there is some evidence that sugary beverages carry a greater risk for metabolic syndrome than when sugar is in the food [17,18]. 

If the speed of ingestion (which would translate into differences in delivery time to the liver) is important in how fructose causes metabolic effects, then the ideal study to test the hypothesis would be to give a fructose-containing solution at two different rates to volunteers. We, therefore, performed a study in which apple juice (rich in fructose) was administered over 5 min versus over 1 h to healthy adult volunteers. We compared the metabolic effects of fast and slow fructose consumption by evaluating changes in serum glucose, insulin, homeostatic model assessment (HOMA-IR) score, copeptin, sodium, blood urea nitrogen (BUN), osmolarity, fibroblast growth factor 21 (FGF-21), uric acid, lactate, and phosphate concentrations after fructose ingestion. Serum copeptin is a stable biomarker of vasopressin (AVP) and predicts incident obesity, type II diabetes mellitus (DM) [19,20], and kidney and CVD [21]. In addition, recent studies showed that fructose ingestion increases serum copeptin in both animals and humans [22,23,24], and that blocking the vasopressin 1b receptor can block the metabolic effects of fructose [24]. FGF-21 is a key regulatory hormone in glucose and lipid metabolism which was shown to enhance hepatic fatty acid oxidation and ketogenesis in both animals and humans, whereas serum uric acid is a critical factor in the pathogenesis of fructose-induced metabolic syndrome and a strong predictor of clinical outcome [25]. We hypothesized that slow consumption of soft drinks would cause less change in serum copeptin, sodium, BUN, osmolarity, FGF-21, lactic acid, uric acid, and phosphate compared to fast consumption, which may be helpful to alleviate the metabolic effects of excess soft-drink consumption. 

## 2. Materials and Methods

### 2.1. Characteristics of the Study Population

Thirty healthy, nonobese participants without any systemic disease and receiving no medications and alcohol were included to this study. All volunteers underwent the intervention at 8:00 a.m. after overnight fasting. Participants did not eat or drink except water for 8 h before the intervention. 

Baseline laboratory tests were performed before starting to drink apple juice. Baseline measurements included serum glucose, insulin, HOMA-IR score, FGF-21, copeptin, osmolarity, sodium, BUN, lactic acid, uric acid, and phosphate, as well as complete blood count, liver, and kidney function tests. The study protocol is summarized in Figure 1. 

The Koc University School of Medicine ethics committee approved the study protocol. Written informed consent was taken from all participants before enrollment.

### 2.2. Study Protocol

Each subject was given a total of 500 mL of 100% pure, commercially available apple juice to consume during the intervention (https://www.isrctn.com/ISRCTN14798840). The participants were randomly allocated to either group 1 or group 2. We used 100% apple juice as the fructose source since it has a higher fructose-to-glucose ratio and has no antioxidant effect [5]. Each participant drank the same apple juice. The participants in group 1 were instructed to drink 125 mL every 15 min under direct supervision four times until a total of 500 mL was consumed. The participants in group 2 were instructed to drink 500 mL of apple juice within 5 min under direct supervision. No other eating or drinking was allowed during the 2 h study period for both groups. The subjects did not perform any physical activity during the study period. 

### 2.3. Serum Measurements

Blood samples were collected for laboratory analysis at baseline and 15, 30, 60, and 120 min after apple juice consumption initiation. For the biochemical analysis, participants’ blood samples were collected into dry tubes and centrifuged promptly (3500× *g*) for 10 min at 4 °C. Sera were separated in aliquots and were frozen rapidly at −80 °C for the copeptin and FGF-21 measurements. The detections of glucose, insulin, sodium, BUN, lactic acid, uric acid, and phosphate were performed immediately using a Roche Cobas 6000 analyzer (Roche, Mannheim, Germany). Serum osmolarity was evaluated using the freezing-point Osmometer K-7400S (Knauer, Berlin, Germany), which permits freezing-point depression to be assessed. Copeptin and FGF-21 measurements in serum were determined by sandwich enzyme-linked immunosorbent assay (ELISA) using commercial kits (Cloud-Clone Corp., Wuhan, China). Intra- and inter-coefficients of variabilities were <10% and <12% for both copeptin and FGF-21 levels. Insulin resistance was estimated according to the HOMA-IR. For calculations, the following formula was used: HOMA-IR = (plasma glucose × plasma insulin/405), where glucose is in mass units (mg/dL) [26]. 

### 2.4. Statistical Analysis

Continuous variables were presented as the mean ± standard deviation (SD), while categorical variables were presented as the frequency (%). Repeated-measures ANOVA was used to examine the difference of variables through time points, as well as the relationship between the time effect and group membership. Continuous variables were examined for normality using the Shapiro–Wilk test. The Mann–Whitney U test and Kruskal–Wallis test were performed to compare two independent continuous variables not normally distributed. All statistics were analyzed using SPSS IBM Corp. Released 2019. IBM SPSS Statistics for Windows, Version 26.0. Armonk, NY: IBM Corp. A *p*-value < 0.05 was considered statistically significant. 

## 3. Results

### 3.1. Characteristics of the Study Population

A total of 30 subjects (13 men and 17 women) participated in the study. Table 1 presents the detailed demographic and clinical data of participants enrolled. The mean age of the subjects was 26.1 *±* 3.7 years. All baseline characteristics were similar between the two groups, including blood pressure, body mass index, tobacco use, and laboratory values.

The baseline plasma glucose, insulin, HOMA-IR score, FGF-21, osmolarity, copeptin, sodium, BUN, uric acid, lactate, and phosphate levels were not statistically different between the groups. Changes in all parameters were significantly different between group 1 and group 2 with the exception of uric acid levels (Table 2).

### 3.2. Effect of Intervention on Serum Glucose, Insulin, and HOMA-IR

In terms of magnitude, the increase in serum glucose was higher and steeper with fast ingestion of the apple juice when compared to that after slow ingestion (Figure 2). In group 2, glucose levels reached a peak after 30 min and declined afterward, while glucose levels demonstrated a subtle incline until 60 min without a dramatic peak level in group 1 (Figure 2A). The increases in serum glucose levels from baseline to 15 min and 30 min were significantly higher in group 2 (*p* = 0.001 for both) (Table 3), while the changes in serum glucose levels after 60 min and 120 min were not statistically different between the two groups.

Serum insulin levels followed a similar trend to serum glucose levels (Figure 2B). Insulin levels showed a sharper and higher increase and an earlier drop after fast ingestion when compared with slow ingestion. Group 1 reached peak insulin levels after 60 min, while group 2 reached peak levels after 30 min. Compared with baseline, the increase in insulin levels was significantly greater in group 2 after 15 min, 30 min, and 60 min (*p* = 0.004, 0.03, and 0.02, respectively; Table 3). As compared with baseline levels, the difference in serum insulin levels remained significantly higher in group 2 until 60 min (Figure 2B).

The shifts in HOMA-IR levels paralleled serum insulin levels (Figure 2C) and increased following juice ingestion in both groups. The HOMA-IR levels of group 2 had a higher rise and quicker decline compared to group 1. The increases in HOMA-IR levels from baseline to 15 min and 30 min were significantly higher in group 2 (*p* = 0.003 and 0.04, respectively; Table 3), while the difference was insignificant after 60 min (Table 3).

### 3.3. Effect on FGF-21

The pattern of serum FGF-21 levels over time was greater and steeper rise after fast apple juice consumption compared to slow consumption. Serum FGF-21 steadily increased in both groups until reaching a peak after 60 min and decreased after 120 min (Figure 2D). The differences in FGF-21 concentrations compared with baseline were consistently higher in group 2 at all times (*p* < 0.01 for all) (Table 3).

### 3.4. Effect of on Serum Copeptin, Sodium, BUN, and Osmolarity

Serum copeptin, sodium, BUN, and osmolarity significantly increased after ingestion of apple juice, and all patterns demonstrated a greater and sharper increase in group 2 compared to group 1. Copeptin concentration achieved its peak after 30 min in both groups and remained markedly higher after fast ingestion compared to slow ingestion (Figure 3A). The change in copeptin concentrations compared with baseline was significantly greater in group 2 after 15 min and 30 min (*p* = 0.005 and *p* = 0.001, respectively), but did not remain significant after 60 min and 120 min (Table 3). Serum sodium, BUN, and osmolarities showed very similar trends with time. In both groups, serum sodium, BUN, and osmolarity started to increase after 15 min (Figure 3B–D). While the changes were statistically significant in both groups, the serum sodium, BUN, and osmolarity levels in group 2 demonstrated dramatic upsurges compared to the modest increases in group 1. Group 2 showed greater increases serum sodium, BUN, and osmolarity after 15 min (0.002, <0.001, and 0.001, respectively), 30 min (<0.001 for all), and 60 min (0.01, 0.003, and 0.04, respectively) (Table 3).

### 3.5. Effect on Plasma Lactic Acid, Uric Acid, and Phosphate

Lactic acid levels had a steep rise starting after 15 min and increased from 1 to 2.6 mmol/L after 30 min in group 2 (Figure 4A). On the other hand, only a slight increase from 0.9 to 1.5 mmol/L was seen after 30 min in group 1. The differences in changes of serum lactic acid compared to baseline were significant after 15 min, 30 min, and 60 min (*p* = 0.001, 0.001, and 0.04, respectively) (Table 3).

After ingestion of apple juice, serum phosphate levels slightly increased in group 2 but did not show a significant change in group 1 (Figure 4B). Compared with baseline, the mean phosphate concentrations were significantly higher in group 2 at all times (Figure 4B), while phosphate levels changed minimally in group 1 (Table 3).

Serum uric acid levels showed a significant difference in pattern between the two groups (*p* < 0.01), whereas the changes in uric acid levels over time were not statistically significant (*p* = 0.4) (Table 2). Although serum uric acid levels in group 2 were higher at all times (Figure 4C), the differences compared with baseline were not significant (Table 3).

## 4. Discussion

Here, we investigated whether the speed of consumption of apple juice would change key modulators of glucose/lipid metabolism and markers of cardiorenal morbidity. Following prior studies suggesting that fructose causes its metabolic effects by decreasing ATP levels in the liver, we hypothesized that the administration of fructose over a shorter time would have greater metabolic effects than giving the same amount over a longer time. Consistent with our hypothesis, we showed significantly greater changes in serum glucose, insulin, HOMA-IR score, FGF-21, osmolarity, copeptin, sodium, BUN, and lactic acid levels with fast intake of apple juice compared to slow intake.

Our primary finding was that the metabolic effects of fructose were much more prominent with fast ingestion of the apple juice. According to the results of a recent study in mice, when fructose is ingested in low amounts, the majority of fructose metabolism occurs in the intestines and only a small remaining portion reaches the liver. However, high amounts of fructose overwhelm intestinal absorption capacity, allowing the liver to have a greater role in its metabolism into glucose, lactate, and glycerate. Either way, fructose metabolism into glucose, lactate, and fatty acids [2] leads to an increase in blood glucose and lactate levels. Our results showed that fast ingestion of 100% apple juice caused a significantly steeper and higher increase in both serum glucose and lactate levels, confirming that fast ingestion of fruit juice causes rapid absorption and metabolization of fructose. It is also plausible that faster ingestion of 100% juice with its high fructose load may saturate intestinal metabolism and cause more fructose-induced liver injury, although the present findings are insufficient to demonstrate this definitively.

One key finding of the present study was that fast ingestion of 100% apple juice resulted in significantly higher FGF-21 levels. FGF-21 is a metabolic hormone synthesized largely by the liver, and fructose is known to be a robust stimulus for FGF-21 release into circulation [27]. It was previously shown that acute ingestion of an oral fructose load results in an acute rise in serum FGF21 levels and a return to baseline within 5 h [28]. In addition to corroborating the previous findings, our results imply that faster ingestion of a fructose content leads to a stronger FGF-21 response. It is thought that FGF-21 may have a protective role in energy metabolism by acting as an insulin sensitizer [29] and by stimulating the oxidation of fatty acids and inhibition of lipogenesis [30]. Furthermore, FGF-21 was shown to protect against fructose-induced liver inflammation in mice [31]. Nevertheless, elevated circulating FGF-21 levels have been associated with poor metabolic health in humans and shown to correlate positively with fasting glucose [32], fasting insulin, and triglycerides and negatively with high-density lipoprotein cholesterol (HDL) [33]. Importantly, high serum FGF-21 is a predictor of obesity, insulin resistance, and metabolic syndrome [33,34]. Although the role of FGF-21 in the context of increased fructose consumption is yet to be uncovered, the increase in FGF-21 after fast juice consumption can be hypothetically explained by a defense mechanism which counterbalances the increased glucose and insulin levels.

Another finding of our study was significantly higher serum osmolarity in response to fast juice consumption. One simple explanation is that this was due to the osmolarity of the fructose that was absorbed. However, we noted an increase in serum sodium that would not be expected if this were the only mechanism. We previously reported that laboratory rats administered chronically fructose solutions also develop high plasma and urine osmolarity in association with a rise in serum copeptin levels [35,36,37]. In these earlier studies, it was shown that the rise in serum vasopressin was not dependent on the osmolarity of the fructose but rather mediated by fructose metabolism [22,24,38]. The mechanism appears to result from a shift of water into the cell, likely from the rapid synthesis of glycogen that carries water [35]. Intracellular fluid shift results in relative hyperosmolarity of plasma. Possibly by inducing faster glycogen synthesis, fast consumption of 100% apple juice may lead to a greater water shift with higher osmolarity as depicted by our results. It can also be hypothesized that ingestion of high amounts of sweetened beverages may cause transitory dehydration. Beverages high in carbohydrate content are also hypertonic and greatly increase the secretion of fluid into the gut, which slows fluid absorption and uptake into the body [39]. The rise in the osmolarity, in return, directly stimulates copeptin secretion, which is depicted by the elevated copeptin levels. In addition, fructose is known to stimulate the production of vasopressin in the hypothalamus [22]. The stimulation of the V1b receptor by vasopressin has been found to be responsible for many of the metabolic features mediated by fructose in mice. While the mechanism could involve stimulation of ACTH or glucagon, in the reported study, the primary finding was that V1b receptor amplified fructokinase expression, an effect that can enhance the production of fat and other features of the metabolic syndrome. We postulated that this may represent another function of vasopressin to protect against dehydration, as fat can act as a source of metabolic water (due to the ability of fat to release water upon its oxidation) [24].

Our results further depicted a striking increase in serum BUN levels after rapid juice ingestion. Previous studies have proposed that fructose-rich diet is a risk factor for the development and progression of CKD due to enhanced glomerulosclerosis and tubulointerstitial inflammation [40]. Fructose is predominantly transported by GLUT5 and GLUT2 in the proximal tubule, where it is phosphorylated by fructokinase (ketohexokinase), which causes ATP consumption and uric acid generation through xanthine oxidoreductase [40]. Therefore, fructose is an acute source of oxidative stress and uric acid production in the kidneys [41]. In rat models, fructose-containing sweetened beverages were found to markedly aggravate the renal impairment induced by mild heat dehydration [36]. Acute effects of fructose load on renal clearance are yet to be elucidated. Nevertheless, the acute and temporary increase in BUN parallels the increase in serum sodium. This finding points to fluid shift as the main mechanism of the present changes in serum BUN.

In our study, serum osmolarity rose, most likely as a result of the temporary hypernatremia induced by intracellular water shift, increased BUN, and, to a lesser extent, the higher plasma glucose levels observed with rapid apple juice ingestion. Although the change in osmolarity was short-lived and not clinically significant in the present study, it should be noted that hyperosmolarity was shown to be associated with increased mortality in patients with acute coronary syndrome [42,43], acute kidney injury [44], and hypertension [45,46]. Previous studies have also shown an association between hyperosmolarity and obesity [47]. Elevations in copeptin (a stable biomarker of vasopressin) predict the development of metabolic syndrome and are present in most subjects with this condition [48,49]. Our results showed dramatically and significantly higher concentrations of copeptin with fast consumption of 100% juice in the first 30 min. As discussed earlier, the rise in vasopressin levels is now known to be a direct mediator of fructose-induced metabolic syndrome, consistent with the hypothesis that rapid ingestion carries greater metabolic consequences.

Our findings also suggest that 100% apple juice significantly increases plasma phosphate levels. During intrahepatic fructose metabolism, fructose phosphorylation depletes the intracellular phosphate, which results in conversion of adenosine triphosphate (ATP) to adenosine monophosphate (AMP), which is later catabolized into uric acid [50]. Indeed, previous studies found decreased plasma inorganic phosphate levels in response to fructose [51]. It is highly likely that the elevated levels of serum inorganic phosphate after apple juice ingestion were related to the phosphorus content of apple juice itself rather than to fructose metabolism, as apple juice is known to be a rich source of phosphorus, containing 7 mg of phosphorus per 100 mL [52].

Lastly, we observed increases in uric acid; however, differences between uric acid concentrations between fast and slow ingestion were not statistically significant. Fructose is the only sugar that increases purine degradation and production of uric acid [53]. Importantly, hyperuricemia has a major causal role in fructose induced metabolic syndrome [54], likely due to inhibition of insulin-mediated endothelial nitric oxide synthesis [55,56]. Given that uric acid mediates many hazardous effects of fructose, limiting the rise in uric acid after the ingestion of fructose may have clinical benefits in the long term. Our findings showed a consistent trend toward higher uric acid levels with fast ingestion, while the association was found statistically insignificant. However, the present study may be underpowered to detect small differences.

This study had some limitations. First, the sample size was small, which limited the ability to detect slight differences. Secondly, we assumed that the observed changes were mainly the response to the fructose content of 100% apple juice. By preferring apple juice over citrus juice, which has minimal known antioxidant activity, we tried to limit such a confounding impact. Nevertheless, we used 100% apple juice instead of a fructose-containing solution to mimic the “real-life” consumption of a common beverage. Other ingredients of apple juice may have directly or indirectly affected the biomarkers evaluated. It is known that copeptin level is higher in males compared to females; thus, the additional male subject in group 2 may explain the slightly higher copeptin level at baseline. However, the relative increase in copeptin in group 2 compared to group 1 suggests that the responses to the apple juice based on speed of ingestion were significant. Moreover, we did not control the water intake during the 8 h period of fasting before the experiments, and differences in water intake between groups could have impacted the copeptin levels, especially if consumed immediately before the commencement of the experimental protocol. It should also be noted that current findings are insufficient to conclude any long-term consequences of fast versus slow fructose ingestion, and future research is needed to elucidate whether such an association exists.

In conclusion, rapid consumption of 100% apple juice causes a more dramatic metabolic response, characterized by increases in serum glucose, insulin, FGF-21, copeptin, sodium, BUN, osmolarity, and lactic acid. The study suggests that the metabolic effects of fructose are not just driven by the amount ingested, but by how rapidly it is ingested. Sugary beverages may be distinct from solid foods in being able to induce more severe metabolic effects due to the frequently rapid speed of ingestion. In clinical practice, advocating the consumption of sweetened beverages and fruit juices in spaced-out smaller portions may alleviate the harmful effects of fructose.

## Figures and Tables

**Figure 1 nutrients-13-01916-f001:**
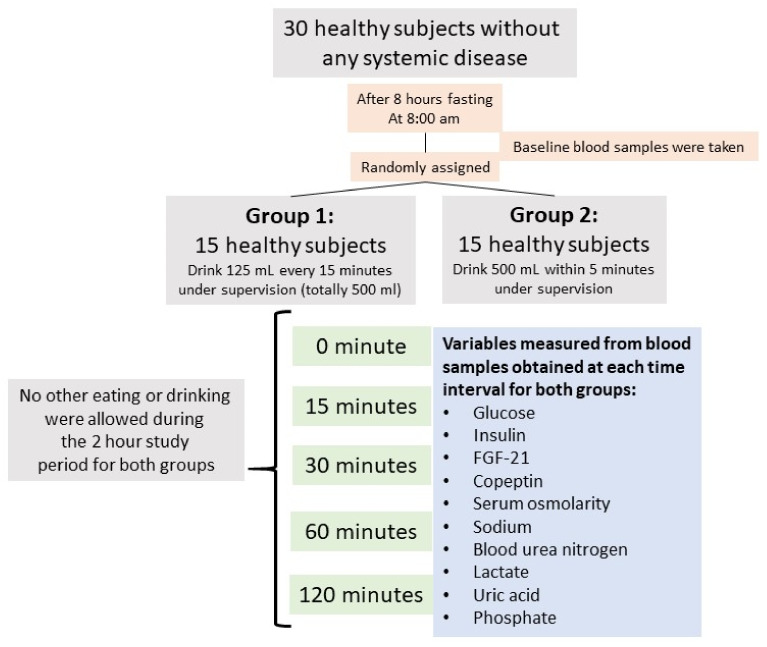
Flow diagram of the study design.

**Figure 2 nutrients-13-01916-f002:**
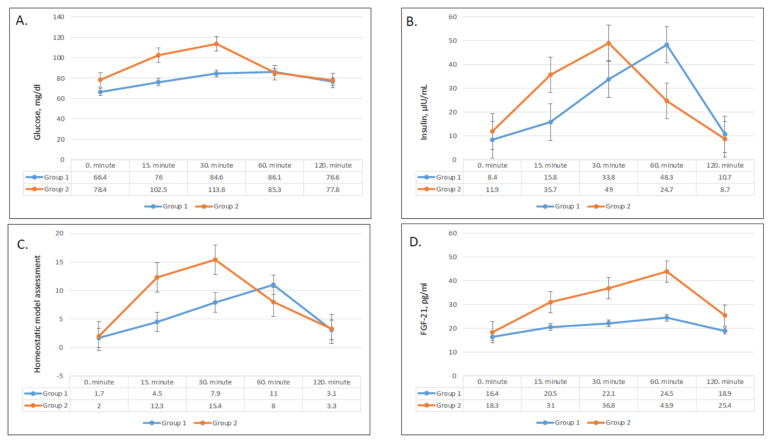
Changes in plasma glucose (**A**), plasma insulin (**B**), HOMA-IR (homeostatic model assessment) (**C**), and FGF-21 (fibroblast growth factor 21) (**D**) values in group 1 and group 2.

**Figure 3 nutrients-13-01916-f003:**
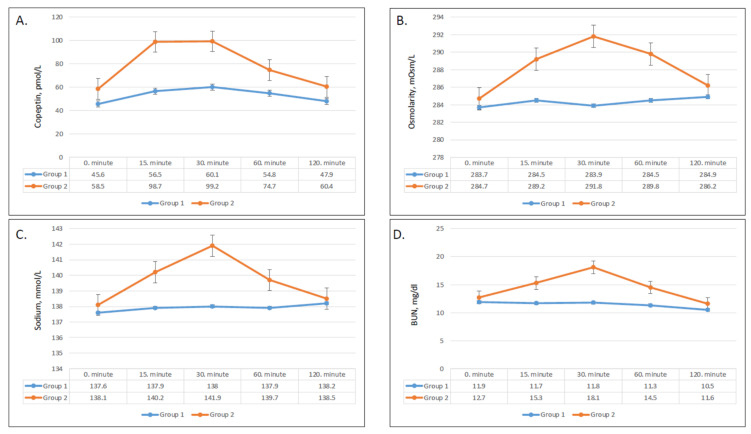
Changes in plasma copeptin (**A**), plasma osmolarity (**B**), sodium (**C**), and BUN (blood urea nitrogen) (**D**) values in group 1 and group 2.

**Figure 4 nutrients-13-01916-f004:**
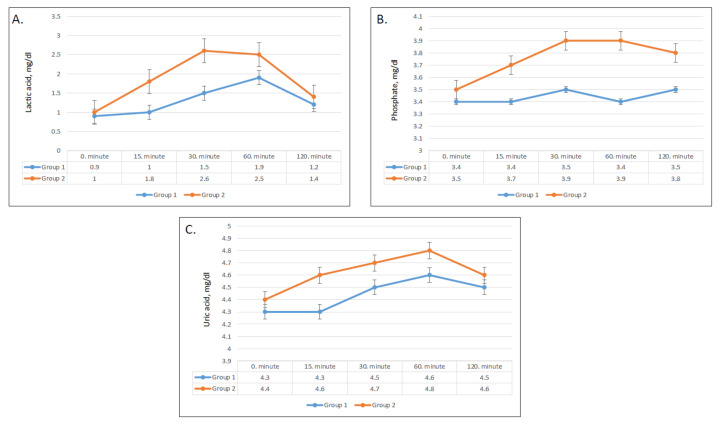
Changes in plasma lactic acid (**A**), plasma uric acid (**B**), and phosphate (**C**) values in group 1 and group 2.

**Table 1 nutrients-13-01916-t001:** Baseline characteristics of the study population.

Characteristics	30 Healthy Subjects			
	Overall	Group 1	Group 2	*p*
Number	30	15	15	
Age, years (mean ± SD)	26.1 *±* 3.7	25.33 ± 1.8	26.9 ± 4.9	0.7
Gender (male) n (%)	13 (43.3%)	6 (40%)	7 (46.7%)	0.9
Smoking, n (%)	4 (13.3%)	2 (13.3%)	2 (13.3%)	0.9
BMI, kg/m^2^ (median ± IQR)	22.9 *±* 4.1	22.86 ± 6.3	22.9 ± 2.1	0.8
Systolic BP (mmHg)	109.2 (105.9 *±* 113.3)	108.5 (104.8 ± 112.2)	109.5 (105.8 ± 113.2)	0.9
Diastolic BP (mmHg)	74.2 (70.9 ± 76.3)	74.5 (70.8 ± 76.8)	73.9 (70.9 ± 75.9)	0.9
Hemoglobin (median ± IQR)	14.4 *±* 3.1	14.1 ± 2.5	14.6 ± 3.1	0.5
ALT-U/L (median ± IQR)	12.9 *±* 11.7	11 ± 9.2	14.3 ± 22.7	0.1
GGT-U/L (median ± IQR)	12 *±* 10	12 ± 8	12 ± 13	0.8
Creatinine-mg/dL (median ± IQR)	0.7 *±* 0.3	0.7 ± 0.2	0.8 ± 0.3	0.5

*p* < 0.05 is significant. BMI—body mass index, BP—blood pressure, IQR—interquartile range, ALT—alanine aminotransferase, GGT—gamma-glutamyl transferase.

**Table 2 nutrients-13-01916-t002:** Serum glucose, insulin, HOMA-IR (homeostatic model assessment) score, FGF-21 (fibroblast growth factor), copeptin, osmolarity, sodium, BUN (blood urea nitrogen), lactate, uric acid, and phosphate levels after the ingestion of 500 mL of apple juice over time.

	Baseline	15 min	30 min	60 min	120 min	*p*-Value ^a^	*p*-Value ^b^
Glucose (mg/dL)						<0.001	<0.001
Group 1	66.4 ± 6.3	76 ± 18	84.6 ± 10	86.1 ± 1.5	76.6 ± 15.1		
Group 2	78 ± 18.7	102.5 ± 17.3	113.6 ± 25	85.3 ± 30.2	77.8 ± 9.2		
Insulin (μIU/mL)						<0.001	0.001
Group 1	8.4 ± 5.5	15.8 ± 10.5	33.8 ± 21.7	48.3 ± 35.8	10.7 ± 5.9		
Group 2	11.9 ± 7.4	35.7 ± 19.4	49 ± 36.2	24.7 ± 24.4	8.7 ± 7.4		
HOMA-IR score						<0.001	<0.001
Group 1	1.7 ± 1.0	4.5 ± 4.0	7.9 ± 3.9	11 ± 5.7	3.1 ± 4.4		
Group 2	2.0 ± 1.0	12.3 ± 9.6	15.4 ± 10.9	8.0 ± 4.1	3.3 ± 2.2		
FGF-21 (pg/mL)						<0.001	<0.001
Group 1	16.4 ± 17.2	20.5 ± 13.7	22.1 ± 16.2	24.5 ± 15.6	18.9 ± 14.1		
Group 2	18.3 ± 15.4	31 ± 12.7	36.8 ± 11.2	43.9 ± 17.1	25.4 ± 17.2		
Copeptin (pmol/L)						<0.001	0.044
Group 1	45.6 ± 52.9	56.5 ±46.8	60.1 ± 52.4	54.8 ± 53	47.9 ± 35.9		
Group 2	58.5 ± 61.8	98.7 ± 64.6	99.2 ± 57.6	74.7 ± 60.9	60.4 ± 30.2		
Osmolarity (mOsm/L)						<0.001	<0.001
Group 1	283.7 ± 5.6	284.5 ± 8.7	283.9 ± 6.5	284.5 ± 6.6	284.9 ± 7.9		
Group 2	284.7 ± 5.4	289.2 ± 6.6	291.8 ± 4.9	289.8 ± 6.8	286.2 ± 5.3		
Sodium (mmol/L)						<0.001	<0.001
Group 1	137.6 ± 1.8	137.9 ± 2.0	138 ± 1.6	137.9 ± 1.5	138.2 ± 1.7		
Group 2	138.1 ± 1.2	140.2 ± 1.0	141.9 ± 0.8	139.7 ± 0.8	138.5 ± 1.2		
BUN (mg/dL)						<0.001	<0.001
Group 1	11.9 ± 4	11.7 ± 4.0	11.8 ± 3.9	11.3 ± 3.6	10.5 ± 3.2		
Group 2	12.7 ± 4.0	15.3 ± 3.4	18.1 ± 4.0	14.5 ± 3.2	11.6 ± 3.5		
Lactic acid (mmol/L)						<0.001	<0.001
Group 1	0.9 ± 0.5	1 ± 0.4	1.5 ± 0.5	1.9 ± 0.7	1.2 ± 0.4		
Group 2	1 ± 0.4	1.8 ± 0.4	2.6 ± 0.8	2.5 ± 0.8	1.4 ± 0.7		
Phosphate (mg/dL)						0.022	<0.001
Group 1	3.4 ± 0.8	3.4 ± 0.7	3.5 ± 0.6	3.4 ± 0.7	3.5 ± 0.8		
Group 2	3.5 ± 0.4	3.7 ± 0.2	3.9 ± 0.3	3.9 ± 0.7	3.8 ± 0.7		
Uric acid (mg/dL)						<0.001	0.48
Group 1	4.3 ± 1.5	4.3 ± 1.4	4.5 ± 1.7	4.6 ± 1.7	4.5 ± 1.7		
Group 2	4.4 ± 1.1	4.6 ± 1.1	4.7 ± 0.9	4.8 ± 1.1	4.6 ± 1.3		

Data are presented as the mean (95% CI) at baseline and least-squares mean (95% CI) at 15, 30, 60, and 120 min. Analysis was conducted using a mixed model for repeated measures, adjusting for baseline values. ^a^
*p-*Values for effect trend over time in all arms (repeated-measures ANOVA). ^b^ *p-*Values for treatment × time interaction, evaluated if changes in Group 1 were different from changes in Group 2 (repeated-measures ANOVA).

**Table 3 nutrients-13-01916-t003:** The differences in glucose, insulin, HOMA-IR (homeostatic model assessment) score, FGF-21 (fibroblast growth factor), copeptin, osmolarity, sodium, BUN (blood urea nitrogen), lactate, uric acid, and phosphate levels compared with baseline after 500 mL of juice ingestion.

	Baseline–15 min	Baseline–30 min	Baseline–60 min	Baseline–120 min
Glucose (mg/dL)				
Group 1	11.6 ± 9.2	20.5 ± 7.9	19.7 ± 13.7	8.3 ± 112.7
Group 2	31.7 ± 14.7	38.2 ± 13.4	13.6 ± 15.7	6.7 ± 8.3
*p-*Value	<0.001	<0.001	0.213	0.983
Insulin (μIU/mL)				
Group 1	11.4 ± 13.5	22.9 ± 12.8	33.9 ± 20.1	6.1 ± 20.2
Group 2	35.8 ± 31.3	42.9 ± 30.2	18.6 ± 15.4	1.9 ± 6.8
*p-*Value	0.004	0.033	0.019	0.494
HOMA-IR (Homeostatic model assessment)				
Group 1	2.9 ± 3.7	6.2 ± 3.6	9.4 ± 6.1	1.5 ± 4.6
Group 2	10.3 ± 9.8	13.4 ± 10.7	6.0 ± 4.2	1.2 ± 1.8
*p-*Value	0.003	0.040	0.078	0.494
FGF-21 (pg/mL)				
Group 1	3.7 ± 7.1	6.6 ± 7.5	6.5 ± 6.9	−1.3 ± 8.7
Group 2	14.8 ± 7.5	21.9 ± 7.4	24.5 ± 11.7	8.9 ± 8.4
*p-*Value	<0.001	<0.001	<0.001	0.001
Copeptin (pmol/L)				
Group 1	18.8 ± 30.6	23.3 ± 36.4	20.3 ± 31.9	2.6 ± 10.6
Group 2	35.2 ± 24.1	46.2 ± 28	20.2 ± 24.2	2.7 ± 15.4
*p*-Value	0.005	0.001	0.576	0.604
Osmolarity (mOsm/L)				
Group 1	0.5 ± 2.5	1.2 ± 1.5	1.3 ± 1.3	0.8 ± 2
Group 2	5.1 ± 2.3	6.4 ± 2.1	4.3 ± 3.7	1.5 ± 2.3
*p*-Value	<0.001	<0.001	0.044	0.494
Sodium (mmol/L)				
Group 1	2.9 ± 3.7	6.2 ± 3.6	9.4 ± 6.1	1.5 ± 4.6
Group 2	10.3 ± 9.8	13.4 ± 10.7	6.0 ± 4.2	1.2 ± 1.8
*p-*Value	0.002	<0.001	0.013	0.656
BUN (mg/dL)				
Group 1	0.3 ± 1.7	0.4 ± 1.4	0.3 ± 1.2	0.6 ± 1.7
Group 2	2.1 ± 0.9	3.8 ± 1.0	1.6 ± 1.2	0.3 ± 1.2
*p-*Value	<0.001	<0.001	0.003	0.827
Lactic acid (mmol/L)				
Group 1	0.1 ± 0.3	0.6 ± 0.5	1.1 ± 0.6	0.5 ± 0.7
Group 2	0.7 ± 0.3	1.5 ± 0.5	1.5 ± 0.5	1.5 ± 0.4
*p*-Value	<0.001	<0.001	0.04	0.351
Phosphate (mg/dL)				
Group 1	−0.1 ± 0.2	−0.1 ± 0.1	−0.2 ± 0.3	0.04 ± 0.5
Group 2	0.1 ± 0.1	0.3 ± 0.2	0.4 ± 0.3	0.3 ± 0.2
*p-*Value	0.001	<0.001	<0.001	0.026
Uric acid (mg/dL)				
Group 1	0.1 ± 0.1	0.2 ± 0.3	0.3 ± 0.3	0.3 ± 0.3
Group 2	0.1 ± 0.2	0.3 ± 0.9	0.4 ± 0.3	0.2 ± 0.3
*p-*Value	0.931	0.308	0.411	0.675

## Data Availability

Data described in the manuscript will be made available upon request pending approval and payment if necessary.

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
