# Peer review of "The Speed of Ingestion of a Sugary Beverage Has an Effect on the Acute Metabolic Response to Fructose"

_nutrients, 2021, doi:10.3390/nu13061916_

Round 1

Reviewer 1 Report

Thank you for the possibility to review the manuscript entitled "The Speed of Ingestion of a Sugary Beverage Has an Effect on the Acute Metabolic Response to Fructose" by Mehmet Kanbay et al, an experimental study in humans randomly assign to either fast or slower intake of apple juice after 8 hours fasting. The authors present data showing that fast ingestion of apple juice causes greater metabolic response than slower ingestion. The comprehensive tables and figures contribute to the presentation of the results. The design of the study is straight forward and the article is easy to follow and the majority is well-written.

Major comments

  1. The levels of copeptin are not comparable with copeptin concentrations in previous studies. Despite an overnight fast it is surprising to find mean concentrations that are 10-fold higher than one would expect in healthy young individuals. I would prefer to re-analyse the samples with another sandwich elisa method to verify that the results are correct. Or, is it possible that it is in fact a calculation error, that it is actually not 45 pmol/L and 58 pmol/L but 4,5 pmol/L and 5,8 pmol/L in group 1 and 2 respectively at baseline.
  2. Copeptin is generally significantly higher within men than women, and usually sex-specific analyses are preferred. The low numbers are therefore a problem in this study. The fact that there is one more male subject in group 2 is also problematic. Please discuss.
  3. In row 97 it is stated that “Participants did not eat or drink except water for 8 hours before the intervention”. Was the water intake somehow controlled, or were they allowed to drink as much as they wanted within 8h before the intervention? It would have been preferred to control for water intake as copeptin is rapidly and easily suppressed by water intake. Especially since the discussion focuses on possible water shifts. Please discuss.
  4. It should be more clear in the limitation section that it is not possible to claim any negative long-term consequences from fast ingestion of fructose based on the current results.
  5. It is stated in the discussion (line 295) that “Acute effects of fructose load on renal clearance are yet to be elucidated, nevertheless, the acute and temporary increase in BUN parallel with the increase in serum sodium points to fluid shift as the main mechanism of the present changes in serum BUN”. Furthermore it is stated in the discussion (line 277) that the authors previously found that rats administered chronically fructose solutions develop high plasma and urine osmolarity. Why was not urine samples taken at baseline and after the intervention? It would have been interesting to follow for example urine osmolality to better understand possible fluid shifts.
  6. The authors state (line 309) that “the rise in vasopressin levels is now known to be a direct mediator of fructose-induced metabolic syndrome, consistent with the hypothesis that rapid ingestion carries greater metabolic consequences.” It would be interesting to explore this hypothesis further. As the rapid fructose load seems to lead to a higher copeptin which activates the vasopressin 1b receptor, which not only stimulates the production of fat but also would be expected to increase ACTH and cortisol concentrations as well as glucagon, it would be very informative to additionally include these analyses in the manuscript (i.e. ACTH, cortisol, glucagon).
  7. In the discussion, it is rather surprising that the, in my point of view, most apparent explanation to increased copeptin concentration is not discussed: elevated serum osmolality (the nr 1 stimulus for vasopressin secretion). It is difficult to claim that the rise in vasopressin is directly mediated by fructose when there is in fact an increase in plasma osmolality (suggested to be mediated by a shift of water) that parallels the increase in plasma copeptin. Instead, the authors highlights and discuss the fact that fructose stimulates the production of vasopressin in the hypothalamus (which is an interesting aspect and indeed deserves to be mentioned, as this mechanism may be expected to increase any osmosensor driven vasopressin secretion).

Minor comments:

  1. Well-written introduction but should preferably be slightly shortened.
  2. In row 115 it is stated that the participants in group 2 were instructed to drink the 500 ml of apple juice within 5 minutes under direct supervision. In figure 1 it is stated that the participants in group 1 were assigned to the fast ingestion. Please correct the figure.
  3. It is stated that continuous variables were examined for normality by using the Shapiro Wilks test. Usually copeptin is not normally distributed and therefore usually given as median (25-75 percentile). The large CI would perhaps also support a non-normal distribution.
  4. Please make sure that numbers in y-axis are visible in Figure 3B and 4A.
  5. I would prefer if concentrations were given as mmol/L in addition to mg/dl (at least in text).
  6. In tables, Lactic acid is given as mmol/L but in figure it is said to be given as mg/dl.
  7. I would like the authors to refer to the current study as “ the study “ and not “ these studies “ (line 342)
  8. Serum osmolality, not osmolarity, is the recommended unit.
  9. In row 90-92 it is written that “We hypothesized that slow consumption of soft drinks would cause less change in serum copeptin, sodium, BUN, osmolarity, FGF-21, lactic acid, uric acid and phosphate which may be helpful to alleviate the health effects of excess soft drink consumption.” I would like to include “compared to fast consumption” in the sentence.
  10. Info about fruit juices is found in the discussion (rows 242-251) and would probably suit better in the background and method section.
  11. In figure 1, it should be groups, not group.
  12. In row 282 it is written that “Beverages high in carbohydrates content are also hypertonic and greatly increase the secretion of fluid blood into the gut therefore slowing the fluid uptake by the body”. Please rephrase to make the sentence more comprehensive. Should it be “..increase the secretion of fluid into the gut…”?

Author Response

Comment 1: The levels of copeptin are not comparable with copeptin concentrations in previous studies. Despite an overnight fast it is surprising to find mean concentrations that are 10-fold higher than one would expect in healthy young individuals. I would prefer to re-analyse the samples with another sandwich elisa method to verify that the results are correct. Or, is it possible that it is in fact a calculation error, that it is actually not 45 pmol/L and 58 pmol/L but 4,5 pmol/L and 5,8 pmol/L in group 1 and 2 respectively at baseline.

Response 1: Thank you for this comment. The ELISA kit we used is considered by the manufacturer to be sensitive, specific and with good intra- and inter-assay variation. However, it is not the standard Thermofischer copeptin kit used by many groups. Baseline copeptin levels will vary with country and hydration status and might be expected to be higher in warmer countries, especially in fasting states. Our baseline levels may seem slightly high but a recent study using the Thermofischer assay reported mean copeptin levels of 12 pmol/l without fasting (Sofia Enhorning, The Journal of Clinical Endocrinology & Metabolism, Volume 104, Issue 6, June 2019).  Our measurements were done after overnight fasting. Importantly, we have different groups, so the copeptin measurements can be compared between groups and also longitudinally for specific individuals. This showed that the data grouped consistently and suggests that the general interpretation for differences should be accurate.  Unfortunately, we are unable to run the Thermofischer assay.

Comment 2: Copeptin is generally significantly higher within men than women, and usually sex-specific analyses are preferred. The low numbers are therefore a problem in this study. The fact that there is one more male subject in group 2 is also problematic. Please discuss.

Response 2: Thank you for this instructive suggestion. Due to the small sample size, it is hard to do such an analysis. We mentioned this as a limitation in the discussion section. However, what we are interested in is not so much the difference between groups, but the respective rise in copeptin levels within a group.  As can be seen in the Table, the relative rise in copeptin levels is dramatically different between groups with a much greater rise in copeptin in Group 2.  Hence, we feel the data truly shows a difference in copeptin responses depending on the speed of the apple juice ingestion.

(Line 380-384)

It is known that copeptin level is higher in males compared to females, and there was one more male subjects in group 2 of which may explain the slightly higher copeptin level at baseline. However, the relative increase in copeptin in Group 2 compared to Group 1 suggests that the responses to the apple juice based on speed of ingestion were significant.’

Comment 3: In row 97 it is stated that “Participants did not eat or drink except water for 8 hours before the intervention”. Was the water intake somehow controlled, or were they allowed to drink as much as they wanted within 8h before the intervention? It would have been preferred to control for water intake as copeptin is rapidly and easily suppressed by water intake. Especially since the discussion focuses on possible water shifts. Please discuss.

Response 3: We thank the reviewer for this important suggestion. Although we did not control it, we agree that controlling the water intake during the 8 hours of fasting would have been valuable for ensuring the reliability of copeptin levels. We have raised the issue in the limitations section as follows;

(line 384-387)

“Moreover, we did not control the water intake during the 8 hour period of fasting before the experiments and differences in water intake between groups could have impacted the copeptin levels, especially if consumed immediately before the commencement of the experimental protocol.”

Comment 4: It should be more clear in the limitation section that it is not possible to claim any negative long-term consequences from fast ingestion of fructose based on the current results.

Response 4: Thank you for this suggestion, we have included this statement in the limitation section of the revised manuscript.

(Line 387-389)

“It should also be noted that current findings are insufficient to conclude any long-term consequences of fast versus slow fructose ingestion and future research is needed to elucidate whether such associations exists.” 

Comment 5: It is stated in the discussion (line 295) that “Acute effects of fructose load on renal clearance are yet to be elucidated, nevertheless, the acute and temporary increase in BUN parallel with the increase in serum sodium points to fluid shift as the main mechanism of the present changes in serum BUN”. Furthermore it is stated in the discussion (line 277) that the authors previously found that rats administered chronically fructose solutions develop high plasma and urine osmolarity. Why was not urine samples taken at baseline and after the intervention? It would have been interesting to follow for example urine osmolality to better understand possible fluid shifts.

Response 5: We thank the reviewer for this very constructive suggestion and agree that assessment of urine samples would have been very useful to deepen our understanding of the water shifts. While designing the study, we unfortunately did not plan to collect urine samples, but we regard this recommendation very helpful for our future studies.

Comment 6: The authors state (line 309) that “the rise in vasopressin levels is now known to be a direct mediator of fructose-induced metabolic syndrome, consistent with the hypothesis that rapid ingestion carries greater metabolic consequences.” It would be interesting to explore this hypothesis further. As the rapid fructose load seems to lead to a higher copeptin which activates the vasopressin 1b receptor, which not only stimulates the production of fat but also would be expected to increase ACTH and cortisol concentrations as well as glucagon, it would be very informative to additionally include these analyses in the manuscript (i.e. ACTH, cortisol, glucagon).

Response 6: Thank you for this interesting suggestion. Regarding the activation of vasopressin 1n receptor, assessing the effects of the rapid fructose load on ACTH and glucagon would have been of great interest. Unfortunately, we could not analyze the ACTH and glucagon levels in the current study. We do wish to follow-up with a future study that includes these measurements.

Comment 7: In the discussion, it is rather surprising that the, in my point of view, most apparent explanation to increased copeptin concentration is not discussed: elevated serum osmolality (the nr 1 stimulus for vasopressin secretion). It is difficult to claim that the rise in vasopressin is directly mediated by fructose when there is in fact an increase in plasma osmolality (suggested to be mediated by a shift of water) that parallels the increase in plasma copeptin. Instead, the authors highlights and discuss the fact that fructose stimulates the production of vasopressin in the hypothalamus (which is an interesting aspect and indeed deserves to be mentioned, as this mechanism may be expected to increase any osmosensor driven vasopressin secretion).

Response 7:  We appreciate this point. There have been very good studies that show that fructose stimulates vasopressin independent of osmolality. Specifically, a paper by Wolf et al showed that infusion of fructose intravenously in humans increases vasopressin levels while isoosmolar glucose did not[1].  More recently, we published a paper showing that fructose stimulates vasopressin in hypothalamic explants but not in explants from mice lacking fructokinase (fructokinase knockout) despite similar osmolality[2].  We have now made this clear.  While we cannot exclude a role for osmolality in the rise of copeptin with rapid fructose ingestion, this would not have expected to raise serum Na.  Indeed, the data is consistent with our prior studies showing that fructose stimulates vasopressin but causes a shift in plasma water to the intracellular compartment resulting in increased osmolality[3].

  In the revised manuscript we have added these comments (Line 301-307)

“One simple explanation is that this was due to the osmolality of the fructose that was absorbed. However, we noted an increase in serum sodium that would not be expected if this were the only mechanism. We have previously reported that the laboratory rats administered chronically fructose solutions also develop high plasma and urine osmolarity in association with a rise in serum copeptin levels [3-5]. In these earlier studies it was shown that the rise in serum vasopressin was not dependent on the osmolality of the fructose but rather was mediated by fructose metabolism [1,2,6].

Minor comments

Comment 1: Well-written introduction but should preferably be slightly shortened.

Response 1: Thank you, we have shortened the introduction as suggested. (Lines 43-44, 46-48, 52-53, 55, 71-74 deleted)

Comment 2: In row 115 it is stated that the participants in group 2 were instructed to drink the 500 ml of apple juice within 5 minutes under direct supervision. In figure 1 it is stated that the participants in group 1 were assigned to the fast ingestion. Please correct the figure.

Response 2: Thank you, we have corrected the error in the figure.

Comment 3: It is stated that continuous variables were examined for normality by using the Shapiro Wilks test. Usually copeptin is not normally distributed and therefore usually given as median (25-75 percentile). The large CI would perhaps also support a non-normal distribution.

Response 3: Thank you for this suggestion. We consulted our statistician again and she (A. Baygul, one of co-author) recommended to keep as it is. If you want, we will be happy to change.

Comment 4: Please make sure that numbers in y-axis are visible in Figure 3B and 4A.

Response 4: Thank you for the notice, we have corrected the figures.  

Comment 5: I would prefer if concentrations were given as mmol/L in addition to mg/dl (at least in text).

Response 5: Lactate concentrations were changed to mmol/L (Lines 218, 220)

Comment 6: In tables, Lactic acid is given as mmol/L but in figure it is said to be given as mg/dl.

Response 6: Thank you, we corrected, it should be mmol/L.

Comment 7: I would like the authors to refer to the current study as “ the study “ and not “ these studies “ (line 342)

Response 7: Thank you, we changed the phrase as suggested.

Comment 8: Serum osmolality, not osmolarity, is the recommended unit.

Response 8: Thank you! We did measure osmolarity and the unit of osmolarity is ‘Osm/L’. We therefore have kept the values as shown.

Comment 9: In row 90-92 it is written that “We hypothesized that slow consumption of soft drinks would cause less change in serum copeptin, sodium, BUN, osmolarity, FGF-21, lactic acid, uric acid and phosphate which may be helpful to alleviate the health effects of excess soft drink consumption.” I would like to include “compared to fast consumption” in the sentence.

Response 9: Thank you, we have changed the sentence as suggested.

(Lines 105-108)

“We hypothesized that slow consumption of soft drinks would cause less change in serum copeptin, sodium, BUN, osmolarity, FGF-21, lactic acid, uric acid and phosphate compared to fast consumption, which may be helpful to alleviate the metabolic effects of excess soft drink consumption.”

Comment 10: Info about fruit juices is found in the discussion (rows 242-251) and would probably suit better in the background and method section.

Response 10: Thank you, we have moved the related part to introduction

Comment 11: In figure 1, it should be groups, not group.

Response 11: Thank you, we have corrected the error (as “both groups”)

Comment 12: In row 282 it is written that “Beverages high in carbohydrates content are also hypertonic and greatly increase the secretion of fluid blood into the gut therefore slowing the fluid uptake by the body”. Please rephrase to make the sentence more comprehensive. Should it be “..increase the secretion of fluid into the gut…”?

Response 12: Thank you for the suggestion, we have rephrased the sentence as suggested.

(Line 313-315)

“Beverages high in carbohydrates content are also hypertonic and greatly increase the secretion of fluid into the gut, which slows fluid absorption and uptake into the body.”

Reviewer 2 Report

The article entitled “The Speed of Ingestion of a Sugary Beverage Has an Effect on the Acute Metabolic Response to Fructose” is interesting. As the authors discuss, the increased and excessive consumption of sugary beverages containing high fructose levels has become a concern, since elevated fructose intake has been correlated with the development of obesity, type 2 Diabetes and other metabolic disorders. A human study based on understanding the effects of a common sugary drink ingestion is of major importance because it highlights what happens in “real life” and can help to make people more aware about the effects of excessive sugar intake.

After revising the manuscript, there are several points that need to be highlighted.

Minor points:

units: minutes- min or x'-should be uniform along the manuscript.

The abbreviations or full text should be uniformed (eg. HOMA-IR or Homeostatic model assessment) for all captions.

page 2, line 68- it is missing ATP in full.

page 2, line 84- DM is not defined.

page 2, line 86 - “and that blocking the vasopressin 1b receptor can block the effects of fructose.”   a  reference is missing at the end of the sentence.

page 2, line 92 - “health effects” is confusing. Do you mean metabolic effects?

The amount of sugar content and the percentage of fructose/glucose in the ingested 100% apple juice should be included, if known.

page 4, line 131: the sentence “ Serum copeptin....vasopressin levels (28)” would better fit on another section as Introduction or Discussion.

Are Systolic BP and Diastolic BP presented in Table 1 as average and as median ± IQR? Please clarify once it is done for the other parameters. Diastolic BP for Group 2 is similar to Systolic BP for the same group. That value needs to be corrected.

Statistical analysis of data presented in Table 1 does not show any (*) on the values although it is mentioned below.

Table 2- Osmolarity values do not show “±”

Table 3- title: “...after 500 ml of juice ingestion” for clarity.

page 4, line149: for simplification could not be just described as 13 men and 17 women?

page 9, line 251: please highlighting that 100% fruit juice the sugar content is higher because it is more concentrated than that in fresh whole fruit where fructose is present in low amounts ( apple  fructose content is about 6%).

page 9, line 252: for clarity, when a result is stated to be higher or lower for one group, it should be compared with that obtained for the other group, eg. “...were much more prominent with fast ingestion of the apple juice when compared with those after slow ingestion”. I suggest this kind of sentence to discuss the results along the manuscript.

page 9, line 262: the statement that “ Acute ingestion of an oral fructose...within five hours” should be highlighted that is a result obtained from another work although a reference had been added. Like “ It was shown that acute ingestion...” would be more clear but it is just a suggestion.

page 10, line 281: “...hypothesized that ingestion of high amounts of...”

page 10, line 296: “...to be elucidated. Nevertheless... increase in BUN parallels...sodium. This finding  points out to a fluid shift...” for clarity.

page 10, line 295-297: please provide a brief description about the fluid shift, for clarity.

page 10, line 305: Does “ hyperosmolarity is a characteristic of subjects with obesity” mean that it is  associated with obesity? Is it present in all obese subjects or it is very common in obese subjects?

page 10, line 309: “100% apple juice” instead “ soft drink”. 100% juice technically is not a sweetened beverage once it does not contain added sugars despite its high sugar content as a result from fruit concentration.

page 10, line 285: “...vasopressin 1b receptor where it stimulates the production of fat...metabolic syndrome”. I suggest to revise this sentence and make it more clear in terms of the stimulation of fat and other features.

page 11, line 330: it should read “statistically not significant”. Although it has not been found a statistically significant association between higher uric acid levels and fast juice ingestion, an association could be found when using a larger sample size.

References need to be formatted in a consistent way.

A figure depicting the results obtained would give value to the manuscript.

Major points:

The authors state that the majority of ingested fructose is metabolized mainly by the liver, however, a recent study in mice using stable isotopes demonstrated that a significant fraction of fructose metabolism occurs in the intestine and the remaining fraction reaches the liver. Since hepatic glucokinase activation only requires a small amount of fructose, this is not incompatible with extensive intestinal metabolism of fructose. However, elevated fructose intake may saturate the capacity of the enterocytes, resulting in a higher fructose amount passing to the liver, leading to a metabolic disease. In the present study, a high fructose content juice was ingested in a fast mode. It could be expected that it exceeds the capacity of the enterocytes in metabolizing fructose which spilled over to the liver. The results obtained and described in the present work are not incompatible with the intestinal fructose metabolism.

page 2, line 86- references 21 and 25 are the same reference.

page 2, line 87- Authors state that FGF21 is a key regulatory hormone in glucose and lipid metabolism.  Could the authors provide a more clear insight about this mechanism?

Figure 1: on the top rectangle, “included” could be avoided. Groups are exchanged: group 1 is group 2   according to the results presented along the manuscript. When describing the measured  variables (blue rectangle), it should be highlighted they were performed in blood samples, as the authors did for baseline.

page 3, line 11: Authors used 100% apple juice as fructose source, however total sugar content and also  fructose/glucose percentages are not mentioned, which in my point of view, would be an important information for better follow the results obtained and described on the Results section.

page 10, line 292: “fructose reuptake in the kidney causing uric acid production with oxidative stress in the proximal tubule”. Could the authors include a brief explanation about fructose reuptake?           

Author Response 

Reviewer 2

Minor points:

  • units: minutes- min or x'-should be uniform along the manuscript.

Response 1: Thank you. We have changed the manuscript to include only “minutes”.

  • The abbreviations or full text should be uniform (eg. HOMA-IR or Homeostatic model assessment) for all captions.

Response 2: Thank you, we have revised the text to include only HOMA-IR and BUN for blood urea nitrogen .

  • Page 2, line 68- it is missing ATP in full.

Response 3: Thank you for the notice, it has been corrected. Line 79, “adenosine triphosphate (ATP)”

  • page 2, line 84- DM is not defined.

Response 4:  Thank you, it has been defined.

Comment 5: page 2, line 86 - “and that blocking the vasopressin 1b receptor can block the effects of fructose.”   a  reference is missing at the end of the sentence

Response 5: Thank you, the reference has been added.

Comment 6: page 2, line 92 - “health effects” is confusing. Do you mean metabolic effects?

Response 6: Thank you for the notice, the phrase has been changed to “metabolic effects”.

 Comment 7: The amount of sugar content and the percentage of fructose/glucose in the ingested 100% apple juice should be included, if known.

Response 7: Thank you for this suggestion. The specific content of fructose and glucose was not specified on the package insert. We asked the company and are awaiting their response. The important aspect is that this is a typical type of juice that is ingested and all subjects received the apple juice from the same vendor.

 Comment 8: page 4, line 131: the sentence “ Serum copeptin....vasopressin levels (28)” would better fit on another section as Introduction or Discussion.

Response 8: Thank you for the suggestion. As there was already a similar sentence in the introduction section, the mentioned sentence has been deleted.

 Comment 9: Are Systolic BP and Diastolic BP presented in Table 1 as average and as median ± IQR? Please clarify once it is done for the other parameters. Diastolic BP for Group 2 is similar to Systolic BP for the same group. That value needs to be corrected.

Response 9: Thank you. It was unintentionally wrongly written. We corrected it.

 Comment 10: Statistical analysis of data presented in Table 1 does not show any (*) on the values although it is mentioned below.

Response 10: We are sorry for the confusion, there is no (*) since none of the values is significant. We have deleted the description about (*) under the table.

 Comment 11: Table 2- Osmolarity values do not show “±”

Response 11: Thank you, this has been added.

 Comment 12: Table 3- title: “...after 500 ml of juice ingestion” for clarity.

Response 12: The title has been changed as suggested.

 Comment 13: page 4, line149: for simplification could not be just described as 13 men and 17 women?

Response 13: The description has ben changed accordingly.

(Line 162) A total of 30 subjects (13 men and 17 women) participated in the study.”

 Comment 14: page 9, line 251: please highlighting that 100% fruit juice the sugar content is higher because it is more concentrated than that in fresh whole fruit where fructose is present in low amounts (apple fructose content is about 6%).

Response 14: Thank you for this instructive suggestion, we have emphasized the point as suggested.

(Line 52-55) “Fructose can vary in percentage from 50 to 65% of sugar content in sugar sweetened beverages and up to 67% in some 100% fruit juices, which is up to 10 times more than the whole fruit.”

 Comment 15: page 9, line 252: for clarity, when a result is stated to be higher or lower for one group, it should be compared with that obtained for the other group, eg. “...were much more prominent with fast ingestion of the apple juice when compared with those after slow ingestion”. I suggest this kind of sentence to discuss the results along the manuscript.

Response 15: Thank you for this suggestion, we have revised the results section to include comparison sentences as suggested.

(Line 183-184) the increase in serum glucose was higher and steeper with fast ingestion of the apple juice when compared with those after slow ingestion”

(Line 199-200) Insulin levels showed a sharper and higher increase and an earlier drop after fast ingestion when compared with slow ingestion.”

(Line 206-207) “The HOMA-IR levels of group 2 had a higher rise and quicker decline compared to group 1”

(Line 211-212) “The pattern of serum FGF-21 levels over time was greater and steeper rise after fast apple juice consumption compared to slow consumption”

(Line 217-220) “Serum copeptin, sodium, BUN and osmolarity significantly increased after ingestion of apple juice and both patterns demonstrated greater and sharper increase in group 2 compared to group 1. Copeptin concentration achieved its peaks after 30’ in both groups and remained markedly higher after fast ingestion compared to slow ingestion”

 Comment 16: page 9, line 262: the statement that “ Acute ingestion of an oral fructose...within five hours” should be highlighted that is a result obtained from another work although a reference had been added. Like “ It was shown that acute ingestion...” would be more clear but it is just a suggestion.

Response 16: The sentence has been changed as suggested.

(Line 285-287) “It was previously shown that acute ingestion  of an oral fructose load results in an acute rise in serum FGF21 levels and a return to baseline within five hours (28).”

 Comment 17: page 10, line 281: “...hypothesized that ingestion of high amounts of...”

Response 17: Sentence has been changed as suggested.

(line 312-313)It can also be hypothesized that ingestion of high amounts of sweetened beverages may cause transitory dehydration.”

 Comment 18: page 10, line 296: “...to be elucidated. Nevertheless... increase in BUN parallels...sodium. This finding  points out to a fluid shift...” for clarity.

Response 18: Thank you for the suggestion, the sentence has been changed as suggested.

(Lines 335-338) “Acute effects of fructose load on renal clearance are yet to be elucidated. Nevertheless, the acute and temporary increase in BUN parallels the increase in serum sodium. This finding points out to fluid shift as the main mechanism of the present changes in serum BUN.”

 Comment 19: page 10, line 295-297: please provide a brief description about the fluid shift, for clarity.

Response 19: A description of the fluid shift has been provided as follows.

(Line 308-312) “The mechanism appears to result from a shift of water into the cell, likely from the rapid synthesis of glycogen that carries water (41). Intracellular fluid shift results in relative hyperosmolarity of plasma. Possibly by inducing faster glycogen synthesis, fast consumption of 100% apple juice may lead to a greater water shift and therefore higher osmolality as depicted by our results.”

 Comment 20: page 10, line 305: Does “ hyperosmolarity is a characteristic of subjects with obesity” mean that it is  associated with obesity? Is it present in all obese subjects or it is very common in obese subjects?

Response 20: Hyperosmolarity, or underhydration, was found to be strongly associated with both obesity and diabetes from an analysis of NHANES[7]. However, we agree that it is not universal and that it is better to refer to it as an association. We have modified the paper as follows:

(Line 345-346) Previous studies also show an association between hyperosmolarity and obesity (46).”  

 Comment 21: page 10, line 309: “100% apple juice” instead “ soft drink”. 100% juice technically is not a sweetened beverage once it does not contain added sugars despite its high sugar content as a result from fruit concentration.

Response 21: Thank you for the notice, the correction has been done.

 Comment 22: page 10, line 285: “...vasopressin 1b receptor where it stimulates the production of fat...metabolic syndrome”. I suggest to revise this sentence and make it more clear in terms of the stimulation of fat and other features.

Response 22: Thank you for the notice, we have revised the sentence to make it more clear.

(Line 318-325) The stimulation of the V1b receptor by vasopressin has been found to be responsible for many of the metabolic features mediated by fructose in the mouse. While the mechanism could involve stimulation of ACTH or glucagon, in the reported study the primary finding was that V1b receptor amplified fructokinase expression, an effect that can enhance the production of fat and other features of the metabolic syndrome. We have postulated that this may represent another function of vasopressin to protect against dehydration, as fat can act as a source of  metabolic water (due to the ability of fat to release water upon its oxidation) (24).

 Comment 23: page 11, line 330: it should read “statistically not significant”. Although it has not been found a statistically significant association between higher uric acid levels and fast juice ingestion, an association could be found when using a larger sample size.

Response 23: Thank you, the sentence has been changed as suggested.

(Lines 363-364) “we observed increases in uric acid, however difference between uric acid concentrations between fast and slow ingestion was statistically not significant.”

 Comment 24: References need to be formatted in a consistent way.

Response 24: We are using Endnote for references and were not aware of the inconsistency. We would be happy to modify any specific errors.

Comment 25: A figure depicting the results obtained would give value to the manuscript.

Response 25: Thank you. We have not done this, since we already have 4 figures and 3 tables. However, if this is considered critical, we would be happy to do this.

Major points:

 Comment 1: The authors state that the majority of ingested fructose is metabolized mainly by the liver, however, a recent study in mice using stable isotopes demonstrated that a significant fraction of fructose metabolism occurs in the intestine and the remaining fraction reaches the liver. Since hepatic glucokinase activation only requires a small amount of fructose, this is not incompatible with extensive intestinal metabolism of fructose. However, elevated fructose intake may saturate the capacity of the enterocytes, resulting in a higher fructose amount passing to the liver, leading to a metabolic disease. In the present study, a high fructose content juice was ingested in a fast mode. It could be expected that it exceeds the capacity of the enterocytes in metabolizing fructose which spilled over to the liver. The results obtained and described in the present work are not incompatible with the intestinal fructose metabolism.

Response 1: Thank you for this very informative suggestion, we have now revised the related parts to include the mentioned findings.

(Line 66-69) “Recent studies suggest that after ingestion in low doses, fructose is primarily metabolized in the small intestines, while high doses saturate the absorption of fructose in the intestine, which causes higher amounts of fructose to be carried and metabolized in the liver”

(Line 269-281) “According to the results of a recent study in mice, when fructose is ingested in low amounts, the majority of fructose metabolism occurs in the intestines and only a small remaining portion reaches the liver. However, high amounts of fructose overwhelm intestinal absorption capacity, in which case liver partakes an important role in its metabolism into glucose, lactate, and glycerate. Either way, fructose is metabolized into glucose, lactate and fatty acids (2) and leads to an increase in blood glucose and lactate levels. Our results showed that fast ingestion of 100% apple juice caused a significantly steeper and higher increase in both serum glucose and lactate levels, confirming that fast ingestion of fruit juice causes rapid absorption and metabolization of fructose. It is also plausible that faster ingestion of 100% juice with its high fructose load may saturate intestinal metabolism and cause more fructose-induced liver injury, although present findings are insufficient to demonstrate such effect.”

Comment 2: page 2, line 86- references 21 and 25 are the same reference.

Response 2: Thank you, we have corrected the error corrected.

Comment 3: page 2, line 87- Authors state that FGF21 is a key regulatory hormone in glucose and lipid metabolism.  Could the authors provide a more clear insight about this mechanism?

Response 3: Thank you for the suggestion, we have expanded the explanation in the revised manuscript.

(Line 101-103) FGF-21 is a key regulatory hormone in glucose and lipid metabolism which was shown to enhance hepatic fatty acid oxidation and ketogenesis in both animals and humans”

 Comment 4: Figure 1: on the top rectangle, “included” could be avoided. Groups are exchanged: group 1 is group 2   according to the results presented along the manuscript. When describing the measured  variables (blue rectangle), it should be highlighted they were performed in blood samples, as the authors did for baseline.

Response 4: Thank you for this constructive suggestion, we have changed Figure 1 as suggested. A small version of the figure is given below.

Comment 5: page 3, line 11: Authors used 100% apple juice as fructose source, however total sugar content and also  fructose/glucose percentages are not mentioned, which in my point of view, would be an important information for better follow the results obtained and described on the Results section.

Response 5: Thank you for this suggestion. The specific content of fructose and glucose was not specified on the package insert. We asked the company and are awaiting their response. The important aspect is that this is a typical type of juice that is ingested and all subjects received the apple juice from the same vendor.

Comment 6: page 10, line 292: “fructose reuptake in the kidney causing uric acid production with oxidative stress in the proximal tubule”. Could the authors include a brief explanation about fructose reuptake?   

Response 6: Thank you, now we have included a brief discussion on fructose reuptake in kidney.

(Line 329-333) “Fructose is predominantly transported by GLUT5 and GLUT2 in the proximal tubule, where it is phosphorylated by fructokinase (ketohexokinase) which causes ATP consumption and uric acid generation through xanthine oxidoreductase (39). Therefore, fructose reuptake is an acute source of oxidative stress and uric acid production in the kidneys (40).”

Round 2

Reviewer 1 Report

Thank you for the revised version of the manuscript entitled "The Speed of Ingestion of a Sugary Beverage Has an Effect on the Acute Metabolic Response to Fructose" by Mehmet Kanbay et al.

The authors have answered point-by-point to my comments. Unfortunately, the manuscript suffers from several limitations including lack of urine samples, lack of measurements of ACTH/cortisol and glucagon and most importantly, abnormal copeptin concentrations. The authors fail to present accurate explanations to the highly elevated concentrations measured in the current study. They refer to concentrations measured in a study by Enhörning et al (JCEM 2019), but in fact the copeptin concentrations presented by Kanbay et al are 4-fold higher than the concentrations presented by Enhörning et al. In contrary to what is stated by Kanbay et al, copeptin is measured in fasting plasma samples in the study by Enhörning et al. Furthermore, the copeptin concentrations in the study by Enhörning et al are expected to be higher than normal since the participants recruited to this particular study were selected from a segment of the population  with the highest fasting plasma copeptin concentrations. Kanbay et al also state that one may expect higher copeptin concentrations in warmer countries. One previous study measured copeptin concentrations in a South-African population and found fasting plasma concentrations of around 4.5 pmol/L (Enhörning et al, Endocrine 2019; 65(2): 304–311). I agree that copeptin can be compared between groups and also longitudinally within specific individuals in the current study, but one cannot completely ignore that the concentrations are abnormally high, and this needs to be addressed somehow, otherwise the reliability of the study will be highly impaired.  

Minor comment regarding Response nr 8; I have already understood that you measured osmolarity, but I would have preferred a re-calculation and the use of osmolality (osm/kg) which is a temperature independent unit. With regard to this comment, please check the consistency of wording concerning the use of osmolarity and osmolality throughout the manuscript (as the wording alternates throughout the manuscript).

Author Response

Dear Editor,

We thank the Reviewers and Editorial Board for their consideration of our revised manuscript. Below, we have provided response to the issues raised by the reviewers and made edits based on their suggestions. We will be happy to address any further queries that arise. We look forward to hearing from you at your earliest opportunity.

Sincerely

Mehmet Kanbay, Richard J. Johnson, Miguel A. Lanaspa on behalf of all coauthors

Reviewer comment:

The authors have answered point-by-point to my comments. Unfortunately, the manuscript suffers from several limitations including lack of urine samples, lack of measurements of ACTH/cortisol and glucagon and most importantly, abnormal copeptin concentrations. The authors fail to present accurate explanations to the highly elevated concentrations measured in the current study. They refer to concentrations measured in a study by Enhörning et al (JCEM 2019), but in fact the copeptin concentrations presented by Kanbay et al are 4-fold higher than the concentrations presented by Enhörning et al. In contrary to what is stated by Kanbay et al, copeptin is measured in fasting plasma samples in the study by Enhörning et al. Furthermore, the copeptin concentrations in the study by Enhörning et al are expected to be higher than normal since the participants recruited to this particular study were selected from a segment of the population  with the highest fasting plasma copeptin concentrations. Kanbay et al also state that one may expect higher copeptin concentrations in warmer countries. One previous study measured copeptin concentrations in a South-African population and found fasting plasma concentrations of around 4.5 pmol/L (Enhörning et al, Endocrine 2019; 65(2): 304–311). I agree that copeptin can be compared between groups and also longitudinally within specific individuals in the current study, but one cannot completely ignore that the concentrations are abnormally high, and this needs to be addressed somehow, otherwise the reliability of the study will be highly impaired. 

Answer: Thank you for your reconsideration. Upon your suggestions, we have rechecked our analysis again and reperformed sandwich enzyme immunoassay for the copeptin measurement (By one of co-author Said Incir). The detection range of our kit was between; 15.6-1,000 pg/mL. The concentration of the standard solutions were; 1.000, 500, 250, 125, 62.5, 31.2, 15.6, respectively. Our standard curve and the R2 value are stated at the following;

Curve Name

Curve Formula

A

B

C

D

R2

Fit F Prob

StdCurve

Y = (A-D)/(1+(X/C)^B) + D

0.00467

1.11

1.57E+03

6.41

1

?????

As seen, our R2  value was 1, and all measured values were within the ELISA kit's detecting range (samples did not require a dilution). Considering our R2 value and that no sample was outside the detection range, we are confident that our test results are reliable, although they were found to be higher than the current literature. One reason of such finding may be related to the time lag between sample collection and sample analysis. All samples were collected and stored in -20 degree Celsius until they were analyzed together, which may account for their consistently higher values. Regarding that the main aim of copeptin analysis in this study was to compare the copeptin levels between the study groups rather than to provide baseline copeptin levels, we believe that our copeptin levels provide a reliable comparison between groups.

Minor comment regarding Response nr 8; I have already understood that you measured osmolarity, but I would have preferred a re-calculation and the use of osmolality (osm/kg) which is a temperature independent unit. With regard to this comment, please check the consistency of wording concerning the use of osmolarity and osmolality throughout the manuscript (as the wording alternates throughout the manuscript).

Answer: Thank you for this important suggestion. We directly measured serum osmolarity and not measured osmolality. Therefore we preferred to use osmolarity.  If you still recommend to use happy to convert by calculating from osmolarity. We corrected osmolality as osmolarity throughout the manuscript.
